# The Safety and Effect of Local Botulinumtoxin A Injections for Long-Term Management of Chronic Pain in Post-Herpetic Neuralgia: Literature Review and Cases Report Treated with Incobotulinumtoxin A

**DOI:** 10.3390/jpm11080758

**Published:** 2021-07-30

**Authors:** Songjin Ri, Anatol Kivi, Jörg Wissel

**Affiliations:** 1Neurology and Psychosomatics at Wittenbergplatz, Ansbacher Strasse 17–19, 10787 Berlin, Germany; joerg.wissel@vivantes.de; 2Department of Neurology, Charité University Hospital (CBS), Hindenburgdamm 30, 12203 Berlin, Germany; 3Department of Neurology, Neurorehabilitation, Vivantes Hospital Spandau, Neue Bergstrasse 6, 13585 Berlin, Germany; anatol.kivi@vivantes.de

**Keywords:** post-herpetic neuralgia, botulinumtoxin, neuropathic pain

## Abstract

There are few reports on the safety and effectiveness of long-term botulinumtoxin A (BoNT A) therapy in severe chronic pain of post-herpetic neuralgia (PHN). The literature was searched with the term “neuropathic pain” and “botulinum” on PubMed (up to 29 February 2020). Pain was assessed with the Visual Analogue Scale (VAS) before and after BoNT A therapy. A total of 10 clinical trials and six case reports including 251 patients with PHN were presented. They showed that BoNT A therapy had significant pain reduction (up to 30–50%) and improvement in quality of life. The effect duration seems to be correlated with BoNT A doses injected per injection site. Intervals between BoNT A injections were 10–14 weeks. No adverse events were reported in cases and clinical studies, even in the two pregnant women, whose babies were healthy. The repeated (≥6 times) intra/subcutaneous injections of incobotulinumtoxin A (Xeomin^®^, Merz Pharmaceuticals, Germany) over the two years of our three cases showed marked pain reduction and no adverse events. Adjunctive local BoNT A injection is a promising option for severe PHN, as a safe and effective therapy in long-term management for chronic neuropathic pain. Its effect size and -duration seem to be depended on the dose of BoNT A injected per each point.

## 1. Introduction

Post-herpetic neuralgia (PHN) is a very painful neuropathic condition, which occurs after nerve injury (e.g., demyelination, loss of axons, small-fiber-degeneration, reorganization in the dorsal horn of the spinal cord, and neuroplastic central changes) due to herpes-zoster-virus infection and is defined as a local neuropathic pain lasting for more than three months following the initial acute zoster infection [1,2]. Frequently, the patient’s complaints are burning pain with rushing pain points and dynamic tactile allodynia. Severe PHN reduces QOL (Quality of Life) and often induces sleep disturbance [3]. Despite even escalated pharmacological treatment regimens including anticonvulsants, antidepressants, opioids, and local therapy with lidocaine or capsaicin, patients frequently suffer from severe side effects (especially in patients older than 50 years of age) lacking relevant pain relief. A widely accepted therapy goal with oral medication and/or local therapy strategies for PHN is to reduce the pain by about 30–50% [4,5,6,7,8].

After the first publication on the effect of BoNT A injection against pain due to dystonia by Brin et al. (1987), many studies showed positive effects on chronic pain, including spasticity-associated pain and neurogenic pain [9,10,11,12,13]. The evidence for the efficacy of BoNT A in neuropathic pain relief in humans was firstly presented by Klein in 2004 [14].

## 2. Mechanisms and Sites of Action

Several mechanisms of pain reduction by BoNT A injections have been discussed in the literature, assuming an inhibitory effect on the release of various inflammation-mediated substances (e.g., substance P, glutamate, calcitonin gene-related peptide). This inhibitory effect is mediated by blocking exocytosis by BoNT A, acting via SNAP-25 cleaving [9,10,11,12,13]. The mechanisms mediating the toxin effects on sensory fibers and nociceptors and on the autonomic system are assumed to be mediated in the same way [15]. It was shown that BoNT A leads to a deactivation of sodium channel conductance in cell cultures of central and peripheral neurons [16]. In addition, BoNT A inhibits afferents to muscle spindles, reduces sympathetic signal transmission, and, at least, leads to µreceptor-mediated pain relief at the spinal level [17,18,19]. In 2008, Antonucci et al. suggested that central effects of peripherally applied BoNT A might be due to retrograde transport of the toxin or, alternatively, due to transcytosis leading to an inhibition of neurotransmitters release onto dorsal horn neurons [20]. Moreover, two recent animal studies showed, that BoNT A diminishes the CCI (chronic constriction injury)-induced level of IL-1β (interleukin-1β) and IL-18 within the spinal cord and/or the dorsal root ganglia and, in parallel, enhances the levels of the anti-nociceptive factors IL-1RA (IL-1-receptor antagonist) and IL-10. The authors suggested that BoNT A not only alters neuronal function but also influences spinal microglial cells [21,22]. However, it is still unclear whether those BoNT A actions are mediated locally or indirectly from distant sites. Another recent in-vitro study by Piotrowska (2017) showed new light on the analgesic effect of BoNT A as they suggested a toll-like receptor (TLR2) mediated inhibition of both intracellular signaling pathways and release of pro-inflammatory substance cultured rat neocortical microglial cells [23]. Inhibitory effects of BoNT A were also found on G-proteins and prostaglandin synthase COX-2 (cyclooxygenase-2), the latter known to activate the proinflammatory cytokine Interleukin-1 (IL-1) [24,25]. On that basis, Rojewska et al. (2018) suggested that BoNT A not only exerts its analgesic action directly neuronal but also indirectly via modulating microglial-astrocytic-neuronal crosstalk on the spinal level. In summary, non-neuronal cells are one of the targets of the pain-modulating BoNT A action and have to be taken into account in the context of neuropathic pain treatment [4]. However, the detailed and widely accepted mechanism of BoNT A’s effects against neuropathic pain remains elusive and is still under debate. The positive effect of BoNT A on PHN at the clinical aspect is nevertheless clearly proven lacking data for long-term efficacy and safety. The local BoNT A treatment of PHN is still an “off-label” option without reimbursement of the drug, at least in Germany and other European countries.

## 3. Results

Out of 354 articles, which were found using the terms “neuropathic pain” and “botulinum” on PubMed, only 16 relevant publications were chosen by analyzing their titles and abstracts. Six case/case-series reports, and ten prospective studies totally included PHN 251 patients (Table 1). The follow-up assessment of BoNT effects ranged between one and six months. The maximum total dosage was found for onabotulinumtoxin A (product name: Botox^®^, Allergan, Madison, NJ, USA) amounted to 300 IU and 5–10 IU per injection site, respectively. In three publications, abobotulinumtoxin A (product name: Dysport^®^, Ipsen, Abingdon, UK) was applied, but two studies described no exact information of the used injection methods [26,27]. One study by Jain (2018) performed a total of 500 IU of abobotulinumtoxin A and 20 IU per injection site [28]. Jain (2017) reported the efficacy and safety of BoNT A treatment of PHN in two cases of pregnant women [26]. Despite a wide range of total and single point dosages used and despite different techniques using intra-and/or subcutaneous injections, the positive effect of BoNT A against PHN has been shown over all the patients, albeit with individually different effect sizes. Except for one case report by Emad et al. (2011) [27], the effect of BoNT A against PHN was statistically significant in all included studies [29,30,31,32,33,34,35,36,37,38,39,40].

### 3.1. Case 1

A 56-year-old male with severe pain due to PHN in the area of the left shoulder and ventral thorax was introduced to our office suffering from acute herpes zoster manifestation two years before. The patient was sent from an anesthesiologist. The patient experienced at the beginning pimples within several hours between the spinal processes of the vertebrae T3–T11, at the medial line of left scapulae, and at manubrium sterni. The patient reported a constant high level of pain of about 7–8 VAS interrupted by severe shooting and electric-like pain peaks of some seconds and 8–9 VAS. This situation was given under regular oral drug treatment. Any other accompanying disease was not known. On the second day no other treatment was introduced but local Zn–ointment by the consulted dermatologist. Later on, the scabbing and crusting of the skin injuries healed up over several weeks, but severe pain, scars, and pigmentation as post-herpetic neuralgia remained. The dermatologic and anesthesiologic treatment strategy of this severe PHN included oral pregabaline up to 450 mg per day, morphine up to 80 mg per day, and, at least, local treatment with capsaicin ointment and only short-term effective lidocaine plasters. Mirtazapine (NaSSA: Noradrenergic and Specific Serotonergic Antidepressant) and citalopram (SSRI) had no positive effect on the pain intensity. Six acupuncture therapy sessions over three weeks also had only little effect on pain intensity (VAS 8–9 to 2–8) but with a constant dosage of 80 mg morphine per day.

After referral, we repeatedly did local intra-/subcutaneous multi-point injections of BoNT A (incobotulinumtoxin) with a dosage per treatment session of 150 IU (IU = international units, dilution in normal saline to 100 IU/2 mL) with up to 70–75 injection points and 2–2.5 IU per injection point every three months. After each injection of a total of 20 injection cycles, the patient reported its effect started in 5–7 days after every injection and a stable pain reduction from VAS 7–8 to VAS 4–5 lasted usually for three weeks of latency. Despite the reduction of effect size afterward, it mostly continued to affect until the 11th week. This effect allows the patient to reduce morphine to 40 mg per day (half of the initial dose) for the following years. Over the whole treatment period of three years, no side effects of the BoNT A therapy were detectable and neither reported by the patient, respectively. The total costs of BoNT A therapy and oral medication showed a slight rise of about 30% in comparing with oral medication before additional BoNT A Therapy.

### 3.2. Case 2

A 77-year-old male was referred to our office by a neurologist with pain management specialization. The patient complained of severe PHN (7–8 of VAS) for more than three years, which could not be adequately managed with local and oral medications. Additionally, he complained of sleep disturbance due to severe pain during the night. There were no other accompanying diseases. At the time of the first visit, the patient was on amitriptyline 75 mg and the local application of lidocaine cream (every six hours). Optionally, he added opioids (tramadol), but without experiencing sufficient effects from opioids and morphine. We applied intra- and subcutaneous incobotulinumtoxin A with a total dosage of 160–200 IU, 4–5 IU × 40–45 injection points (1.5–2 cm distance) into the painful skin areas on the left front and lateral neck region between the chin and the claviculae of the left side (dermatome C3). We have had eight therapy sessions every three months and observed a relief to pain from 7–8 of VAS to 4–5 with significant improvement of his sleeping disturbance. The patient reported that the positive effects of BoNT A lasted about 6–8 weeks, after that it reduced but remained until the 12th week and its effect started about one week after the injection. No side effects of local injection of BoNT A were reported in eight therapy cycles.

### 3.3. Case 3

An 80-year-old male was referred to us by an anesthesiologist/pain specialist. The patient complained about severe PHN (6–7 of VAS) for about two years and three months after the herpes zoster infection. The pain was non-manageable by standard oral or local medications. By the treatment of incobotulinumtoxin A injections into the skin of the painful right scapulae area and right chest with total dosages of 200–300 IU, 7.5–10 IU per site at 30–40 sites on a grid with 1.5–2 cm every 3–4 months (total six sessions). The pain intensity was reduced to 3–4 of VAS from a starting level of 6–7 of VA and it lasted for 8–10 weeks and reduced slowly with time. Oral medications and local anesthetic therapy including pregabaline, amitriptyline continued without any changes. Morphine was not tolerated by the patient, because of side effects. Interestingly, in this patient, the positive effect of each BoNT A injection lasted with a latency of 8–10 weeks on average. There were no side effects of the BoNT A injections. In this patient, a distal polyarthritis of the upper extremities treated with regular oral administration of arrhythmicants, and losartan were known.

## 4. Discussion

Most of the publications included in our literature review showed a significant reduction of pain in PHN and secondary complications from PHN as well, such as sleep disorder, the improvement of QOL (Quality of Life) is well documented [14,26,27,28,29,30,31,32,33,34,35,36,37,38,39,40]. Similarly, we observed a significant and stable reduction of pain without side effects in all of our three cases. The safety of the therapy with BoNT A was shown even in two pregnant women, who delivered healthy babies after PHN management with BoNT A [26]. As we only found are case reports or case series with a small number of patients, the formal level of evidence for BoNT A therapy in PHN is still low. However, the beneficial effect of BoNT A therapy in PHN is supported by the publications, which reported more evidence for BoNT A treatment of neuropathic pain in other neuralgias, complex regional pain syndrome, traumatic nerve injury, and diabetic neuropathic pain [41,42].

Comparing subcutaneous BoNT A (Onabotulinumtoxin A) injections (5 IU per every site) with 0.5% Lidocaine, it has to be pointed out that the BoNT effect starts 3–7 days after injections and lasts for about three months, while lidocaine works for one day only. Similarly, subcutaneous lidocaine injections and local anesthetic therapy (creams, pads) have only a short-time effect on PHN [34]. Oral medication is associated with many more side effects and the risk of subsequent dosage increase, while it often lacks to adequately manage the pain to levels below 5 VAS [4,5]. Therefore, the medical needs and the patient’s wants congruently focus on local highly effective therapies for severe PHN, which can be provided by intra-/subcutaneous BoNT A injections.

### 4.1. Applied Botulinumtoxin A

The majority of reports on PHN treatment with BoNT A applied onabotulinumtoxin A (Botox^®^) (100 IU/2 mL 0.9% NaCl). In a total of 174 patients suffering from PHN, onabotulinumtoxin was injected [14,29,30,31,32,33,34,35,36,37]. Only in two case series within a total of 17 patients published by Emad (2011) [27] and Jain (2017) [26] and one cohort study with 13 patients by Jain 2018 [28], abobotulinumtoxin A (Dysport^®^) was used with different dilution rates of the toxin: 500 IU/4 mL 2% Lidocaine or 500 IU/5 mL 0.9% NaCl. Two case reports by Ruiz (2008) [39] and Li (2015) [40] and one study by Eitner (2017) [38] did not disclose the toxin preparation used for the BoNT A treatment. The study by Emad et al. (2011) [27] using abobotulinumtoxin A found no significant reduction of pain which, however, does not allow for the derivation of the subtype-specific effectivity of BoNT A in PHN (Table 1), but rather might result from other dosages and different injection techniques.

In principle, the specific effect of BoNT A on voluntary muscle contraction or muscle tone in spasticity and cervical dystonia shows dose-depended responses, e.g., higher dosage leads to a more pronounced effect on muscle force/tone reduction. Up to now incobotulinumtoxin A never induced a detectable development of antibody-mediated secondary non-responsiveness to BoNT A. Therefore, it could be argued that long-term repetitive BoNT A therapies with higher cumulative dosages might be more stable with this compound [43]. There were, however, no reports using incobotulinumtoxin A in our literature research. For the first time, we introduced incobotulinumtoxin A for the treatment of PHN in our three cases.

### 4.2. Doses and Injection Technique

The respective single doses for one injection site were different in the different studies and case series ranging from 2.5 IU [32] up to 5 IU [29,30,31,33,34,36,37] for onabotulinumtoxin A (injection grid every 1–2 cm one injection site) and 15–20 IU for abobotulinumtoxin A [27,28] (injection grid every 1–3 cm per injection site). As mentioned above, the effect of pain reduction with a single dose of 15 IU abobotulinumtoxin A per injection site (one site every 10 cm^2^) was not significant [27], even though 20 IU of abobotulinumtoxin A per site with an injection interval of 1 cm showed the significant effect in pain reduction on VAS [28]. Perhaps 15 IU in every 10 cm^2^ for abobotulinumtoxin A was probably too low dose or so wide in distance to develop a significant effect on pain reduction. Higher doses per square cm^2^ were significantly more effective although the maximal volume for injections per site is limited therefore the ratio of dilatation is sometimes considered. The studies and reports showed that 5 IU of onabotulinumtoxin A per injection site tended to have a longer effect than 2.5 IU/injection site. Our first case had a single dose of 2–2.5 IU per injection site with incobotulinumtoxin A in a grid of every 1.5–2 cm, in the second case 4–5 IU per injection site, and in the third case 7.5–10 IU per injection site.

The total doses per session were less than 300 IU for onabotulinumtoxin A in most cases and 500 IU for abobotulinumtoxin A, respectively (Table 1). The total doses of incobotulinumtoxin A in our three cases were between 150 and 300 IU per session.

Our experience has shown that the dose per single injection point is important as it tends to be positively correlated to the effect duration: Maximal effects of three weeks for 2–2.5 IU (Case 1), 6–8 weeks for 4–5 IU (Case 2), and 8–10 weeks for 7.5–10 IU (case 3).

In terms of a valid assessment ability of the BoNT A effects on PHN over the studies evaluated, the different injection techniques have to be taken into account, in that most of the studies did not differentiate between intracutaneous and/or subcutaneous injections. However, it is conceivable, that the properties of different tissues might impact the diffusive ability of applied substances and thereby the local toxin concentration. We denoted if the injections of our cases were intra-or subcutaneous. Theoretically, pure intracutaneous injections make a small pallor in the skin, but it lacks an absolute confirmation that the toxin will not be also partially in subcutaneous tissue (or even beyond that). The effects of BoNT A injections in hyperhidrosis were different with respect to these two injection techniques. A direct comparison between the effects of intracutaneous vs. subcutaneous injections of BoNT A does not exist, but clinically the intracutaneous injection is more painful than the subcutaneous treatment and-thereby volume-limited [44].

### 4.3. Materials and Methods

We used PubMed research for the relevant clinical studies and case/case-series reports using the terms “neuropathic pain” and “botulinum” up to the 29th of February 2020. The article selection was done regarding title and abstract. Additionally to the literature research, we introduce three cases of patients suffering from severe PHN, who got repeatedly intra- and subcutaneous injections with incobotulinumtoxin A (Xeomin^®^). Within a color-marked painful area, the dosage per single injection point was 2–2.5 IU for case 1, 4–5 IU for case 2, and 7.5–10 IU for case 3 (IU = international units, dilution in normal saline to 100 IU/2 mL), respectively. Injection points were about 2 cm apart from each other (Figure 1). The total incobotulinumtoxin A doses per session ranged from 150 IU to 300 IU. We used thin needles of 30 gauge and small tuberculin (1 mL volume) syringes for injection. There was no additional local anesthetic. The pain intensity was evaluated by a visual analog scale (VAS) before and every 10–14 weeks after the BoNT A injections (or before the following injections). All other treatments, e.g., oral medications and additional local anesthetic therapies, were documented and the changes made by the patients were captured.

## 5. Conclusions

Botulinumtoxin A seems to be a good option for long-term management in severe PHN, inducing significant pain reduction (up to 30–50% VAS reduction) for up to 3–4 months per injection cycle. It is difficult to reach such distinct effects by classical oral medication and/or local anesthetic therapies.

Adding intra-/subcutaneous BoNT A injections to standard oral medication is more helpful for pain reduction in severe PHN, especially in cases with non-response or non-tolerance of oral medication. There are no differences in both efficacy and safety of the different available BoNT A products and injection techniques (intra- or subcutaneous), respectively. It seems that a longer duration of pain reduction will be achieved by more injection points per session. In the case of ona- and incobotulinumtoxin A, the doses of 5–10 IU per injection point are recommended. The injection interval can be decided individually however, as in the proven range of BoNT A efficacy between 10 and 14 weeks.

## Figures and Tables

**Figure 1 jpm-11-00758-f001:**
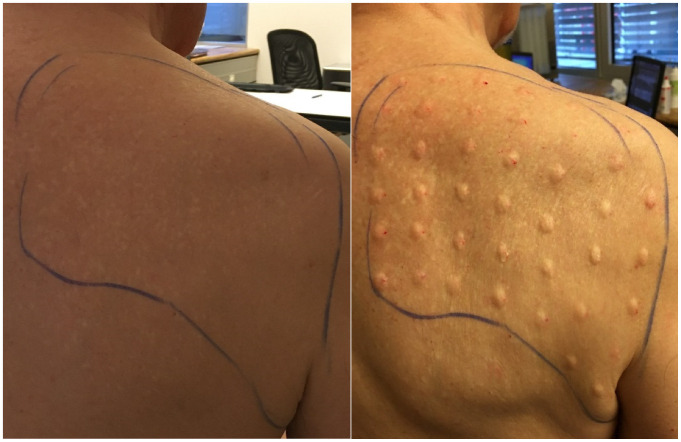
Painful areas marked with blue lines before (**left**) and after (**right**) injections of Incobotulinumtoxin A (case 3).

**Table 1 jpm-11-00758-t001:** Case reports and studies on the management of PHN with BoNT A.

Publications	BoNT A	Study Type	N	I. T.	Doses	Follow-Up	Results
Hu 2020 [29]	Ona A (Chi-Botox)	control Study	33	s.c.	10–20 × 5 IU	16 w	Significant pain reduction (VAS)
Ding 2017 [30]	Ona A (Chi-Botox)	Cohort study	58	s.c.	10–20 × 5 IU	6 m	Significant pain reduction (VAS)
Attal 2016 [31]	Ona A	RCT (total *n* = 66)	5	s.c.	Up to 60 × 5 IU, every 12 w	24 w	Significant pain reduction
Ponce 2013 [32]	Ona A	Cohort Study	12	i.c./s.c.	8–10 × 2.5 IU	3 m	Significant pain reduction
Apalla 2013 [33]	Ona A	RCT	30	s.c.	40 × 5 IU	20 w	greater than 50% pain reduction in VAS, improvement of sleep disorder
Xiao 2010 [34]	Ona A	RCT	60	s.c.	Up to 40 × 5 IU	3 m	Significant pain reduction (mean decreased 4.5 in VAS), improvement of sleep disorder, decreased opioid use
Sotiriou 2009 [35]	Ona A	Case series	3	s.c.	20 × 5 IU	12 w	VAS reduction from 8.3 to 2
Ranoux 2008 [36]	Ona A	RCT (total *n* = 29)	4	i.c.	Up to 40 × 5 IU	14 w	NNT, 50% pain reduction at 12 weeks
Liu 2006 [37]	Ona A	Case report	1	s.c.	20 × 5 IU	52 d	VAS reduction from 10 to 1
Klein 2004 [14]	Ona A	Case report	1	i.c.	20 IU	4 m	Completely pain reduction
Jain 2018 [28]	Abo A	Cohort Study	19	s.c.	25 × 20 IU	16 w	Significant pain reduction (VAS)
Emad 2011 [27]	Abo A	Cohort Study	15	i.c.	15 U/10 cm^2^ (4 mL 2% lidocain)	30 d	pain reduction, but not significant
Jain 2017 [26]	Abo A	Case series (in pregnancy)	2	s.c.	500 IU (5 mL 0.9% NaCl)	16 w	VAS from 9 and 10 to 1
Eitner 2017 [38]	Unknown	RCT, placebo (total *n* = 66)	6	s.c.	Every 1.5–2 cm, 100–300 IU (4 mL 0.9% Nacl)	24 w	Significant pain reduction (VAS)
Ruiz H. 2008 [39]	Unknown	Case report	1	i.c.	unknown	2 m	Dramatically pain reduction
Li 2015 [40]	Unknown	Case report	1	s.c.	100 IU	6 m	Significant improvement in pain relief

N number of subjects with PHN; I. T. Injection technique; N number of patients included; AboA Abobotulinum toxin A; i.c. intracutaneous; Ona A Onabotulinumtoxin A, RCT randomized control trial; s.c. subcutaneous; VAS visual Analog Scale; NNT number needed to treat; Chi-Botox: Chinese Botox; w: week; m: month; d: day.

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
