# Peer review of "The Safety and Effect of Local Botulinumtoxin A Injections for Long-Term Management of Chronic Pain in Post-Herpetic Neuralgia: Literature Review and Cases Report Treated with Incobotulinumtoxin A"

_jpm, 2021, doi:10.3390/jpm11080758_

Round 1
Reviewer 1 Report
This is an interesting review article describing the effects of local botulinum toxin A (BoNT A) injections on long-term management of chronic pain in post-herpetic neuralgia (PHN). The present results from the reviewed studies and case reports show that intra- or subcutaneous injections of BoNT A resulted in a significant pain reduction (up to 30-50%) for up to 3-4 months per injection cycle. Therefore, based on the presented evidence, BoNT A might be a good option for a long-term management of PHN.
Author Response
Dear Sir,
Thank you so much for your precious time and great comments.
We modified the manuscript after your comments.
Best regards

Reviewer 2 Report
The literature review part is not written well, and the contents of the Results are not sufficient for publication. I recommend the authors to focus on case report with deletion of the literature review part.
Author Response
Dear Sir,
Dear Sir,
Thank you so much for your precious time and great comments.
We tried to modify the manuscript (reversion) after your comments.
Best regards

Reviewer 3 Report
This article reports an analysis of the results present in the literature on the effects of botulinum toxin in the treatment of post-herpetic neuropathic pain, together with three cases reports of patients treated by the authors with Xeomin botulinum toxin. The mix of literature review and case reports might seem unusual in a scientific article, however the analysis of the literature appears complete, as well as the information provided regarding the three cases reported. For these reasons, I consider the work acceptable for publication in JPM. However, before final acceptance, some corrections are necessary, as described below.
As main change, I propose to reorganise the presentation of various documents in Table 1. In my opinion, it would be clearer to the reader if the presentation of documents were organized by collecting them following an order based on either the "type of study" or the type of toxin "BoNT A" (I leave to the authors wich order they prefer). Ordering by "type of study" make more immediate to understand in which type of study an analgesic effect has been seen, while ordering by type of BoNTA could make more immediate to recognize which formulation is most effective.
Minor
- Table 1: standardize the symbols for abobotulinum. It appears as Abo, Abo A, aboA at different points in the table
- Line 85: add bracket in 2018; put dot after [28]. not before
- Line 127, 129, 173: change BoNT in BoNT A
- Line 193: abo not italics
Author Response

(The authors gave the same response as above.)

Round 2
Reviewer 2 Report
I do not feel this manuscript is sufficient for publication.